# Characteristics of Fitness-Related Injuries in The Netherlands: A Descriptive Epidemiological Study

**DOI:** 10.3390/sports10120187

**Published:** 2022-11-22

**Authors:** Ellen Kemler, Lisa Noteboom, Anne-Marie van Beijsterveldt

**Affiliations:** 1Team Sports Injury Prevention, Dutch Consumer Safety Institute (VeiligheidNL), Overschiestraat 65, 1062 XD Amsterdam, The Netherlands; 2Faculty of Behavioral and Movement Sciences, Vrije Universiteit Amsterdam, De Boelelaan 1105, 1081 HV Amsterdam, The Netherlands

**Keywords:** exercise, athletes, injuries, prevention, physical activity, gym-based fitness

## Abstract

Although general information is available, specifically detailed information on gym-based fitness-related injuries in the general recreational fitness population is lacking. The aim of our study was to obtain more insight into injuries occurring as a result of gym-based fitness activities. A descriptive online epidemiological study was conducted in November 2020. The survey was distributed by a market research agency to members of their research panel. A total of 494 Dutch fitness participants aged ≥ 18 years (mean 38.9; 59% male) who had sustained a fitness-related injury in the preceding 12 months were included in the study. Most injuries occurred during strength training, individual cardio exercise, yoga/Pilates, cardio exercise in group lessons, and CrossFit. The shoulder, leg, and knee were the most common injured body parts; 73.1% of the injuries occurred during unsupervised gym-based fitness activities, and 46.2% of the injuries occurred during one specific exercise or when using a specific fitness device: running (e.g., on the treadmill) (22.8%); bench press (11.8%); or squats (9.6%). Overuse or overload (*n* = 119), missteps and sprains (*n* = 48), or an incorrect posture or movement (*n* = 43) were most often mentioned as causes of injury. Conclusions: Most self-reported gym-based fitness-related injuries occur during strength training and individual cardio exercise. Special attention should be given to the shoulder during strength training and to the lower extremities during cardio exercise. Injury prevention interventions should be able to be carried out without supervision.

## 1. Introduction

It is commonly known that being physically active contributes to good health, and can reduce the risk of chronic diseases [1]. An increasingly popular method of being active is engaging in gym-based fitness activities, such as strength training, cardio exercise, CrossFit, and body pump. In 2016, 21 percent of the Dutch inhabitants aged 12–79 (about 3 million people) participated in gym-based fitness activities at least once a week, compared to 12 percent in 2001 [2]. Although being physically active through sports participation is considered beneficial, participation in gym-based fitness activities is associated with the risk of injuries [3]. An annual national health survey in The Netherlands shows that many sports injuries occur as a result of gym-based fitness activities [3]. In 2019, the 860,000 reported injuries sustained during gym-based fitness activities accounted for 16% of all sports injuries in The Netherlands [4].

A sports injury is one of the main reasons people terminate sports participation [5]. This could lead to a loss for the participant of all beneficial (physical and mental) effects of sports participation. Hence, the high number of sports injuries as a result of gym-based fitness activities calls for preventive activities. For the development and application of preventive activities and measures, insight into the injury problem of gym-based fitness and fitness activities is necessary [6]. The current fitness branch is a very flexible one, and is constantly offering new developments and activities. A decade ago, it was possible to roughly categorize gym-based fitness activities in the segments “strength”, “cardio,” and “group classes”. Practically all fitness activities took place indoors at the time. Nowadays, the fitness branch offers many activities, including body pump, fighting fit, calisthenics, and bootcamp. Moreover, gym-based fitness activities more often take place in a public space, and are increasingly focused on group fitness with the associated (small) group training zones [2].

Research on injuries sustained during gym-based fitness activities has focused on injuries in Victorian (Australia) emergency departments [7], Israeli young adults aged 20–35 in sports centres [8], or in specific fitness activities such as CrossFit [9,10,11,12,13,14], powerlifting, and weightlifting [15,16]. Specific fitness activities, such as CrossFit and power- and weightlifting, have been studied in several countries, and these results have recently been summarized in systematic reviews [9,10,15]. In 2009, 42% of young adults in Israeli sport centres reported an injury as a result of exercising [8]. The mean prevalence of injuries in CrossFit was 35.3%, with an incidence rate varying between 0.2 and 18.9 per 1000 h of training. The most injured areas were shoulder (26%), spine (24%), and knee (18%) [10]. The mentioned results are related to age groups, specific gym-based fitness activity, or fitness-related injuries treated at emergency departments, but not to the common fitness population or gym-based fitness activities in general. In order to guide and inform prevention of gym-based fitness related injuries, detailed information on gym-based fitness-related injuries in the general recreational fitness population is needed. Some general information, such as the total number of injuries as a result of gym-based fitness activities, is available in, for example, The Netherlands [4]. However, this information is very superficial. Therefore, the aim of the present study was to gain more insight into the characteristics of gym-based fitness-related injuries, the characteristics of the injured fitness athletes, and the type of gym-based fitness activities performed during injury occurrence.

## 2. Materials and Methods

A descriptive epidemiological study involving gym-based fitness participants in The Netherlands was conducted from 5 November to 1 December 2020 via an online questionnaire. The questionnaire was developed based on components of the Health Survey/Lifestyle Monitor, a Dutch national survey used for data collection in the field of lifestyle-related themes such as smoking, alcohol, exercise, nutrition, and accidents and sports injuries. The online questionnaire was distributed by a market research agency to members of their research panel in The Netherlands. The research panel includes around 55,000 Dutch people (18 years and older) who are willing to participate in research. When joining the research panel, participants give permission for participation in future research. Thus, a separate consent form to participate in this particular study was not needed. The study protocol was assessed by the Medical Ethics Review Committee of the Amsterdam University Medical Centre (W20_469 # 20.519). The Executive Board of the METC declared that no ethics approval was necessary because the Medical Research Involving Human Subjects Act did not apply to our study since only questionnaires were used, and they did not have a reason to believe that the psychological integrity of the test subjects was at stake.

In the first step of the study procedure, panel members were questioned via email regarding the inclusion criteria. Participants of gym-based fitness activities that met the criteria of being aged 18 years or older and having sustained a gym-based fitness-related injury in the preceding 12 months were invited to participate in the study. A gym-based fitness-related injury was defined as any physical symptom sustained by an athlete that occurred during or, in the opinion of the participant, as a result of participating in gym-based fitness activities, and that caused the athlete to terminate the fitness training and/or prevent them from participating in the next fitness training session [17,18,19]. Recorded injuries were self-reported by the participants; there was no clinical confirmation of injury or diagnosis in this study. Fitness activities referred to all types of gym-based fitness activities, including strength/cardio exercise and group classes such as spinning, yoga, and bootcamp. The gym-based fitness activities can be performed indoors, outdoors or at home. In the second step, all included participants were asked to (anonymously) complete a single survey which focused on athlete demographics, as well as on the characteristics and consequences of their most recent gym-based fitness-related injury. Furthermore, we asked the participants whether they believed their injury could have been prevented, and if so, how it could have been prevented.

Demographic characteristics which were measured as continuous variables were expressed as mean and standard error (SE). In case of skewed distribution (skewness −1< or > 1), results are presented as median and interquartile range (IQR) or appropriate categories. Categorical variables are expressed as absolute numbers and percentages. The statistical procedures were performed using SPSS^®^ 25 (IBM^®^, New York, NY, USA).

## 3. Results

The research agency contacted 23,862 panel members (members who participate in relevant fitness activities (small sample), members of the general Dutch audience without preselection, and members with low or middle education level), of which 2786 (12%) opened the link to start with the questions regarding the inclusion criteria. For more detailed information about the flow of participants in this study, see Figure 1.

### 3.1. Fitness Participants

Our sample consisted of 494 athletes who, through self-report, indicated having sustained an injury during gym-based fitness activities in the preceding 12 months. More than half of the injured participants were male (*n* = 290, 59%) and the study population’s mean age was 38.9 years, with ages ranging from 18 to 76. Almost two-thirds of the participants (*n* = 309, 63%) had more than two years of gym-based fitness experience, while 22 percent (*n* = 107) of the sample had started only in the previous 12 months. All demographic characteristics are shown in Table 1.

The majority of the participants (*n* = 330, 67%) performed two or more gym-based fitness activities on a regular basis (on average 2.5/athlete). The top five activities were: strength training (*n* = 255, 52%); cardio exercise, individual (*n* = 224, 45%); yoga/Pilates (*n* = 84, 17%); cardio exercise, group lesson (*n* = 79, 16%); CrossFit (*n* = 76, 15%; see Figure 2).

The fitness centre or gym was the most commonly mentioned training location (*n* = 304, 62%). Moreover, 5 out of 10 athletes always trained without a (personal) trainer or instructor (*n* = 255, 52%). Training indoors without a (personal) trainer or instructor was the most common combination (*n* = 150, 30%) of training location and setting (Table 2). A total of 232 participants (47%) reported that they also performed other sports, with almost 60% (*n* = 139) of these participants considering gym-based fitness activities as their main sport.

### 3.2. Gym-Based Fitness-Related Injuries

Most of the gym-based fitness-related injuries were acute (*n* = 315, 64%). One-third of the injuries were classified by the participants as a recurrent injury (*n* = 168, 34%). Injuries were most frequently located in the lower extremities (*n* = 241, 49%). The most commonly injured body parts were the shoulder (*n* = 92, 19%), leg (*n* = 89, 18%), knee (*n* = 65, 13%), ankle (*n* = 60, 12%), and back or thoracolumbar spine (*n* = 54, 11%) (see Table 3).

Injury location differed per gym-based fitness activity. During strength training, the shoulder was injured most often (*n* = 59, 23%), while during individual cardio exercise, cardio exercise in group lessons, and CrossFit, the leg was the most frequently injured body part (*n* = 47, 21%; *n* = 19, 24%; *n* = 19, 25%, respectively). During yoga/Pilates, fitness athletes most frequently injured their back/spine (*n* = 19, 23%).

The most common injury type was a muscle or tendon injury (*n* = 293, 59%), followed by a joint or ligament injury (*n* = 91, 18%) and a contusion (*n* = 31, 6%).

When injury type and location are combined, a muscle or tendon injury of a lower limb was most common (*n* = 122, 25%), followed by a muscle or tendon injury of the upper limb (*n* = 105, 21%), and a joint or ligament injury of the lower limb (*n* = 65, 13%) (see Table 4).

In order to gain insight into the potential cause of the injury, we asked the participants to describe how their fitness-related injury had occurred. The most common answer was that their injury was associated with overuse or overload (*n* = 119). Participants mentioned using too much weight, executing too many repetitions, performing too many activities, or going too fast. In half of the cases, participants did not specify exactly what kind of overload was associated with their injury (*n* = 61). Other commonly self-reported potential causes were missteps and sprains (*n* = 48), or an incorrect posture or movement (*n* = 43).

Seventy percent (*n* = 345) of the injuries in this study occurred in one of the top five most popular gym-based fitness activities, with strength training and individual cardio exercise being the activities associated with the most injuries (151 and 112, respectively; see Figure 2). Every six out of ten injuries occurred indoors in the fitness centre or gym (*n* = 296). Approximately eight percent of the injuries (*n* = 42, 8.5%) occurred at home. More than seventy percent of the injuries (*n* = 361, 73%) occurred during unsupervised gym-based fitness activities (indoors and outdoors without (personal) trainer or instructor) and at home. According to 46% of the fitness athletes (*n* = 228), their injury occurred during one specific exercise (e.g., squats, leg press, etc.) or when using a specific fitness device (cross-trainer, treadmill, etc.). The top five activities or exercises in which injuries occurred were: (1) running (on a treadmill or during outdoor fitness activities) (*n* = 52, 23%); (2) bench press (*n* = 27, 12%); (3) squats (*n* = 22, 9.6%); (4) push-ups (*n* = 14, 6.1%); and (5) cross-trainer (*n* = 10, 4.4%).

### 3.3. Consequences of Gym-Based Fitness-Related Injuries

Two-thirds of the injured participants contacted a health care provider (*n* = 328, 66%). Half of the injuries were examined by a physiotherapist (*n* = 244, 49%). Over 80% of the injuries (*n* = 409, 83%) caused the participant to be absent from sports, school, work, and/or activities at home. Of the injured participants, 73% (*n* = 361) had to miss one or more gym-based fitness activities.

Fifty-five percent (*n* = 272) of the participants indicated that they were fully recovered from their injury: 35% (*n* = 171) were partly recovered, and 11% (*n* = 52) were not recovered at all at the moment of filling out the questionnaire. Of the participants who sustained their injury within the preceding three months, 45% were fully recovered (*n* = 72); of those who sustained an injury in the preceding 4–6 months, 61% (*n* = 104) were fully recovered; of those with an injury in the preceding 7–9 months, up to 70% (*n* = 64) were recovered, and of those who sustained an injury in the preceding 10–12 months, 45% (*n* = 31) were fully recovered. Aside from the indication of being recovered or not, we asked the participants whether they had returned to their normal gym-based fitness activities. Of the 494 participants, only 39% (*n* = 192) had returned to their normal gym-based fitness activities at the time of this study. Of the participants who had sustained their injury in the preceding three months, 32% had returned to their normal gym-based fitness activities (*n* = 51); of those who sustained an injury in the preceding 4–6 months, 44% (*n* = 75) had returned to their normal gym-based fitness activities; of those with an injury in the preceding 7–9 months, up to 48% (*n* = 44) had returned to their normal gym-based fitness activities, and of those who sustained an injury in the preceding 10–12 months, 32% (*n* = 22) had returned to their normal gym-based fitness activities. Of the gym-based fitness participants, 31% (*n* = 153) had modified their gym-based fitness activities as a result of their injury, while 4.5% (*n* = 22) had to quit their gym-based fitness activities permanently. Of the 494 participants, 100 (20%) still experienced pain after their injury, while 9.5% (*n* = 47) experienced reductions in strength.

### 3.4. Prevention

Sixty percent of the participants (*n* = 294) expected that their injury could have been prevented. Commonly mentioned preventive strategies/activities were for example: training less intensively and/or with a training scheme or a better training scheme, exercising with a correct technique/posture, performing a warm-up, and being more careful during training sessions.

## 4. Discussion

The aim of our study was to obtain more insight into the characteristics of injuries occurring as a result of gym-based fitness activities performed indoors, outdoors, or at home. According to our data, seventy percent of the injuries in this study occurred as a result of one of the top five gym-based fitness activities, i.e., strength training, individual cardio exercise, yoga/Pilates, cardio exercise in a group lesson, and CrossFit. The shoulder, leg, and knee were the most common injured body locations in this study, and there are slight differences in injured body locations per fitness activity.

In The Netherlands, countrywide gym-based fitness injury data are available [4]. In 2019, injuries sustained during gym-based fitness activities (*n* = 860,000) accounted for 16% of all sports injuries. In line with the national data, in our study, the shoulder and knee were frequently injured body locations. According to our data, it appears that shoulder injuries were common in strength training, knee injuries in cardio exercise and back injuries in yoga/Pilates. One potential explanation for this result is that the injured body sites are also the body sites that are typically loaded more heavily during these fitness types. For instance, cardio exercise typically involves running and rowing, which may logically result into more knee injuries. Another potential explanation for the high shoulder injury occurrence in strength training is that the traditionally non-weight bearing shoulder complex is loaded in unfavourable positions during strength exercises, and may be more injury-prone than other body parts [20].

In our study, injured participants of gym-based fitness activities more often contacted a health care provider (*n* = 328, 66%) compared to half of the gym-based fitness participants in national data [4]. Additionally, onset of the injuries was more acute (*n* = 315, 64%) than gradual (*n* = 179, 36%), while in the national data, 53% of the injuries had a gradual onset. The design of the study could have contributed to these two differences, as in our study, injuries sustained by participants in the preceding 12 months were included, while in the national data, only injuries in the previous three months were included. The difference in recall period might have led to recall bias, in which severe injuries are, for example, over-reported within a recall period of 12 months, and minor injuries are under-reported. Another explanation for the difference in medical treatment could be the study period in which the studies were conducted. The national data are data from 2019, while our study was performed after the onset of the COVID-19 pandemic, with limited access to medical care.

Of the 494 fitness participants, 39 percent (*n* = 192) had returned to their normal fitness activities at the time of this study. More recently sustained injuries, and thus with a shorter follow-up period in this study, are likely less often fully cured than injuries sustained 10–12 months previously, and, thus, could impede a return to normal fitness activities. Therefore, we asked the fitness participants how long ago they sustained their injury. Analyses showed that fitness athletes who had sustained an injury 10–12 months ago had not fully returned to sports at a higher rate than fitness athletes who had sustained their injury within the preceding 0–9 months. A possible explanation might be that participants who had been injured 10–12 months ago did report more severe injuries, as only 45% of them were fully recovered compared to 55% on average.

Strength training is the most commonly practised fitness activity in our study. Most injuries also occurred during strength training. These results are in accordance with the results of Gray and Finch (2015), who concluded that 55% of the injuries sustained at fitness facilities and treated in emergency departments are related to strength training (free weight activities). Gray and Finch analysed the specific cause of the injuries sustained [7]. In their study, overexertion or strenuous and/or unnatural movement was an important cause of injuries. Participants in our study also perceived this an important cause of injury. Another important cause, which was not often mentioned in our study was ‘crushed by falling/dropped weights’ [7]. As Gray and Finch analysed injuries treated at emergency departments, and we analysed all injuries sustained by a cross-section of the fitness population, this could explain the different causes found.

According to a recent review, injury aetiologies in resistance training (RT) are multifactorial, and they include the impacts of overuse, short post-exercise recovery periods, poor conditioning in the exercised body areas, frequent use of heavy loads, improper technique in certain exercises, and the abuse of performance- and image-enhancing drugs [21]. Professional supervision and adhering to proper lifting techniques and training habits is required for selected exercises in RT programmes. In our study, several injuries occurred during large, more complex exercises such as the “bench press”, “squat”, and “dead lift”. These exercises are often part of RT programmes. Our fitness participants frequently mentioned that their injuries were caused by a “wrong posture or movement” or could have been prevented by using a “good technique or posture”. Other common causes mentioned by the respondents were overuse or overload (*n* = 119), and missteps and sprains (*n* = 48). Several answers were, unfortunately, too general to provide insight into the cause of injury. We believe and agree that with this type of exercises, supervision of technique and movement is a first step in injury prevention, possibly with the help of a trainer or an automatic camera-based feedback system. The more complex exercises might require more experienced trainers, as the trainers’ experience may play a pivotal role in the trainees’ musculoskeletal pain [22]. In a study by Ahmed et al., (2022), fitness participants trained by trainers with lower experience had more than two times the risk of having pain in different sites (IRR: 2.04, 95% CI: 1.50–2.79). More research is, however, needed as to what experience is required to supervise complex exercises and prevent injuries.

Another form of fitness that was often practised by the respondents of the study on gym-based fitness activities is functional fitness, e.g., CrossFit, calisthenics, bootcamp, and HIIT. Functional fitness aims to improve physical fitness by developing aerobic capacity, muscular strength and endurance, speed, coordination, agility, balance, accuracy, flexibility, and stamina [23]. Although functional fitness might differ from strength training, the most affected body location is, again, the shoulder, followed by the lumbar spine and the knee [24]. As CrossFit is one of the most studied forms of functional fitness, this outcome is no surprise [9]. However, almost regardless of the type of fitness activity performed, the shoulder always seems to be at a higher risk of injury, indicating the absolute need for injury prevention measures for this specific body part.

According to van Mechelen’s sequence of prevention [6], insight into injuries, sports activities, and risk factors is necessary to ultimately develop and implement injury-preventive interventions. Based on our data, which fit into the first quadrant of the sequence of prevention, some clues for prevention and practical implications can be derived. Most injuries occurred during strength training and individual cardio exercises. Special attention should be given to the shoulder during almost all types of gym-based fitness activities, but at least during strength training. A practical implication might be that trainers always supervise (complex) exercises that involve the shoulder, until the trainee is capable of executing the exercise in the correct way. As more than seventy percent of the injuries (*n* = 361, 73%) occurred during unsupervised fitness activities and at home, injury preventive interventions must be able to be carried out without supervision, and/or outside the gym, or supervision should be stimulated. Since supervision is not always possible, less complex exercises should be advised in order to prevent the occurrence of injuries.

The identification of risk factors for sustaining a fitness-related injury is another step in the process of developing an injury-preventive intervention [6]. Studies such as that of Gray and Finch [7] and ours solely included injury data. In both studies, most injuries occurred during strength training. This is, however, a very popular fitness activity. The risk of sustaining an injury during strength training might, thus, be rather low compared to other fitness activities. In order to gain more insight into injury risk and risk factors, in future studies, exposure, e.g., hours of fitness participation, should be included, and characteristics of injured fitness athletes should be compared with healthy fitness athletes.

Some strengths and limitations should be addressed. A strength of this study was the gathering of information on injuries sustained by a cross-section of the fitness population. This information gave insight into the injury problem in fitness and fitness activities, and provided some initial clues for injury preventive activities. Some limitations of this study should be addressed as well. First, information was obtained through self-report, which could have led to patient bias (no medical professional was involved in the diagnosis of or reporting on the injury) [25] and socially desirable responses. A fitness injury was defined as any physical complaint sustained by a fitness participant that occurred during or as a result of a fitness training, and that caused the fitness participant to terminate the fitness training and/or prevent them from participating in the next fitness training session. The injury should have been sustained in the preceding 12 months. Longer recall periods of, e.g., 12 months significantly underestimate the injury rate compared to shorter recall periods. Shorter recall periods (1–3 months) are recommended for injury questionnaires [26], and should be used when calculating the overall non-fatal injury incidence rate [27]. However, longer recall periods (12 months) may be safely used to obtain information on the more severe, but less frequent, injuries [27]. Furthermore, a study on the accuracy of patient recall for self-reported doctor visits showed that both bias and variance were minimised for the 12-month recall period, compared to the 2-week and 3-month recall periods [28]. According to these authors, this result may reflect the telescoping that occurs with shorter recall periods (participants reporting important events that fall outside the period) [28]. Our 12-month recall period could have led to recall bias, as less severe injuries could be under-reported. However, as the aim of this study was to gain insight in the injury problem, and then derive first clues for prevention, we believe that our results are still useful.

Finally, the online selection procedure through a respondent panel may have induced selection and participation bias. In advance, from a small sample of the panel, it was known that they participated in gym-based fitness activities. The online questionnaire was sent to this small sample first. Then, a random sample, representative of the Dutch population, was contacted. At last, the sample was enriched with panel members of a low or middle education level to compensate for any under-reporting of these members. The research agency monitored the responses from the panel members.

Although the research agency attempted to minimise selection bias by using a representative sample, (non-) participation bias still might still occur. Non-participation bias can affect the internal validity of the results in every study. In ours, twelve percent of the contacted panel members opened the link to the screening questions. Although a click-through rate of 12% was rather normal for their panel members according to the research agency (addressed in personal communication with the authors), one small explanation for this percentage was that at the time of the study, the research agency had one panel in which both respondents for quantitative research and for qualitative research were members. Since our study was of a quantitative nature, panel members who only wished to participate in qualitative research would not have opened the link. However, this only accounts for 5–10% of the panel. Unfortunately, it was not possible to compare the non-participants with the participants in, e.g., terms of exercises habits. However, the results of our study seem comparable with national results for several characteristics [4]. Hence, we believe that the effect of participation bias on our internal validity is acceptable.

## 5. Conclusions

In our study, in which we aimed to gain detailed information on fitness-related injuries in the general recreational fitness population, most injuries occurred during strength training and individual cardio exercise. Special attention should be given to the shoulder during strength training, and the lower extremities during cardio exercise. Injury preventive interventions must be able to be carried out without supervision, and/or outside the gym, or supervision should be stimulated. For large, more complex, free exercises such as the “bench press”, “squat”, and “dead lift”, supervision of technique and movement by a trainer or, for example, an automatic camera-based feedback system is a first step towards injury prevention.

## Figures and Tables

**Figure 1 sports-10-00187-f001:**
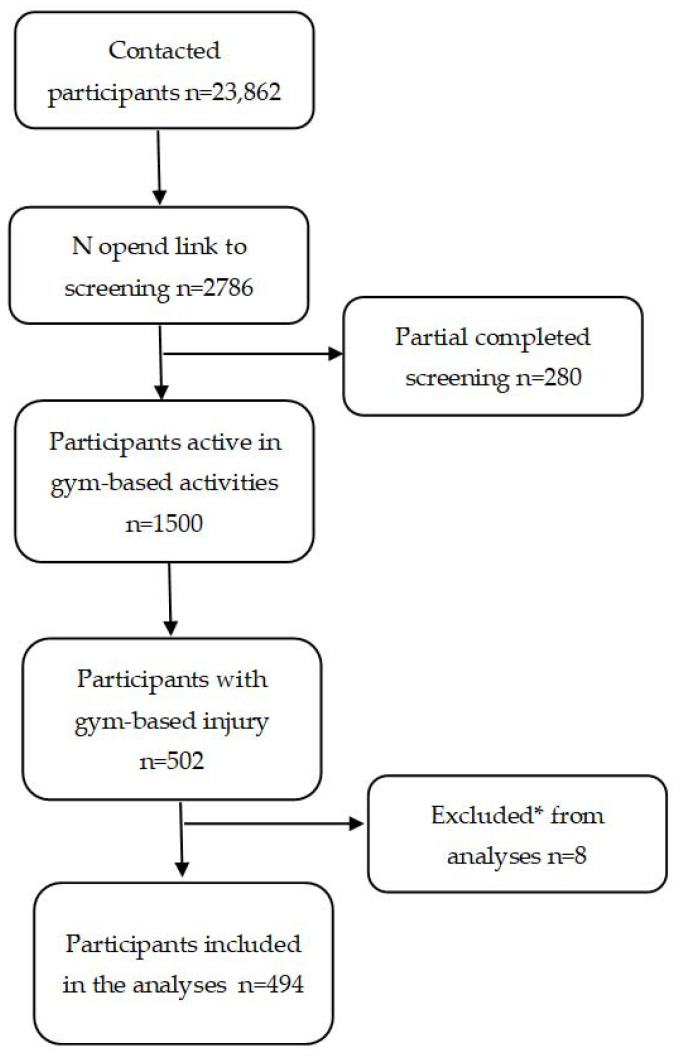
Flow chart of this study. * Respondent activities were not considered to be gym-based fitness activities.

**Figure 2 sports-10-00187-f002:**
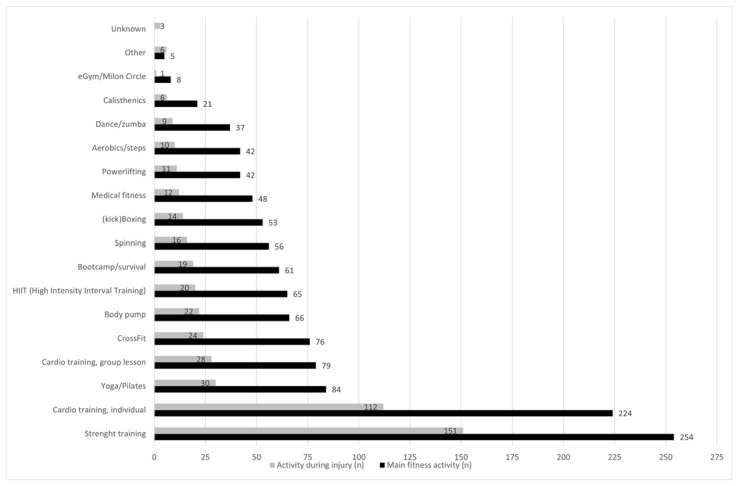
Main fitness activities of the athletes and the activity during injury.

**Table 1 sports-10-00187-t001:** Demographic characteristics of all fitness participants (*n* = 494) and their training details.

Gender, *n* (%)	
Male	290 (59)
Female	203 (41)
Other	1 (0.2)
Age (years), mean (SE; 95% CI)	38.9 (0.597; 37.3–39.6)
Age (years), range (min–max)	58; 18–76
Body height (m), mean (SE; 95% CI)	1.77 (0,415; 176.4–178.1)
Body height (cm), range (min–max)	60; 153–213
Body mass (kg), mean (SE; 95% CI)	78.3 (0.645; 77.2–79.9)
Body mass (kg), range (min–max)	120; 50–170
BMI (height/kg^2^), *n* (%)	
Underweight (<18.5)	10 (2.0)
Normal weight (≥18.5 and <25.0)	276 (56)
Overweight (≥25.0 and <30.0)	165 (33)
Obesity (≥30.0)	40 (8.1)
Fitness training experience (months), *n* (%)	
1–3	12 (2.4)
4–6	30 (6.1)
7–12	65 (13)
13–24	78 (16)
>24	309 (63)
Training frequency per month, *n* (%)	
1–4	120 (24)
5–8	131 (26)
9–12	111 (22)
13–16	68 (14)
17–20	35 (7.1)
>20	29 (5.9)
Training duration per session (minutes), *n* (%)	
0–30	40 (8.1)
31–60	255 (52)
61–90	166 (34)
91–120	27 (5.5)
>120	6 (1.2)
Training location, *n* (%)	
Fitness centre or gym (indoor)	304 (61)
Outdoor	84 (17)
Home	56 (11)
All above options (appr.) equal	50 (10)
Training setting, *n* (%)	
Always without (personal) trainer or instructor	255 (56)
Always with (personal) trainer or instructor	91 (18)
Both with and without (personal) trainer	148 (30)
Working hours per week, mean (SE; 95% CI)	33 (0.544; 31.84–33.97)
Working hours per week, range (min–max)	80 (0–80)
Working hours (h), *n* (%) (*n* = 436)	
<8 h	19 (4.4)
≥8 h and ≤24 h	76 (17)
>24 h and ≤36 h	162 (37)
>36 h	179 (41)
Physical exertion during work, *n* (%)	
Always	23 (4.7)
Often	47 (9.5)
Sometimes	144 (29)
Never	222 (45)

SE = standard error.

**Table 2 sports-10-00187-t002:** Training location and setting.

Location	Setting, *n* (%)			Total, *n* (%)
	Always without (Personal) Trainer or Instructor	Always with (Personal) Trainer or Instructor	Both with and without (Personal) Trainer or Instructor	
Indoor	150 (30)	65 (13)	89 (18)	304 (61)
Outdoor	45 (9.1)	16 (3.2)	23 (4.7)	84 (17)
Home	41 (8.3)	4 (0.8)	11 (2.2)	56 (11)
All three options	19 (3.8)	6 (1.2)	25 (5.1)	50 (10)
Total	255 (52)	91 (18)	148 (30)	494 (100)

**Table 3 sports-10-00187-t003:** Characteristics of all fitness injuries (*n* = 494).

Occurrence (months ago), *n* (%)	
0–3	161 (33)
4–6	172 (35)
7–9	92 (19)
10–12	69 (14)
Setting, *n* (%)	
Indoor, without (personal) trainer or instructor	199 (40)
Indoor, with (personal) trainer or instructor	92 (19)
Outdoor, without (personal) trainer or instructor	120 (24)
Outdoor, with (personal) trainer or instructor	41 (8.3)
At home	42 (8.5)
Location, *n* (%)	
All head/neck injuries	10 (2.0)
Brain	1 (0.2)
Neck or cervical spine	9 (1.8)
All upper limb injuries	147 (30)
Arm or elbow	31 (6.3)
Shoulder	92 (19)
Wrist	14 (2.8)
Clavicle	1 (0.2)
Hand	4 (0.8)
Fingers or thumb	5 (1.0)
All trunk injuries	81 (16)
Back or thoracolumbar spine	54 (11)
Trunk, ribs, abdomen, or organs	9 (1.8)
Hip or pelvis	18 (3.6)
All lower limb injuries	241 (49)
Leg	89 (18)
Knee	65 (13)
Ankle	60 (12)
Foot	26 (5.3)
Toe(s)	1 (0.2)
Total body or more locations	4 (0.8)
Other	11 (2.2)
Type, *n* (%)	
Muscle or tendon injury (e.g., strain)	293 (59)
Joint or ligament injury (e.g., sprain)	91 (18)
Contusion	31 (6.3)
Cartilage damage	11 (2.2)
Fracture or bone stress	10 (2.0)
Wound or abrasion	8 (1.6)
Concussion	1 (0.2)
Other	11 (2.2)
Unknown	38 (7.7)

**Table 4 sports-10-00187-t004:** Cross-tabulation of injury location and type (*n* = 494).

Type	Injury Location, *n* (%)					Total, *n* (%)
	Head/Neck	Upper Limb	Trunk	Lower Limb	Other	
Muscle or tendon injury	5 (1.0)	105 (21)	51 (10)	122 (25)	10 (2.0)	293 (59)
Joint or ligament injury	1 (0.2)	15 (3.0)	9 (1.8)	65 (13)	1 (0.2)	91 (18)
Contusion	0	8 (1.6)	6 (1.2)	17 (3.4)	0	31 (6.3)
Cartilage damage	0	2 (0.4)	1 (0.2)	8 (1.6)	0	11 (2.2)
Fracture or bone stress	0	6 (1.2)	0	2 (0.4)	2 (0.4)	10 (2.0)
Wound or abrasion	0	2 (0.4)	0	6 (1.2)	0	8 (1.6)
Concussion	1 (0.2)	0	0	0	0	1 (0.2)
Other	1 (0.2)	3 (0.6)	5 (1.0)	2 (0.4)	0	11 (2.2)
Unknown	2 (0.4)	6 (1.2)	9 (1.8)	19 (3.8)	2 (0.4)	38 (7.7)
Total	10 (2.0)	147 (30)	81 (16)	241 (49)	15 (3.0)	494 (100)

## Data Availability

The datasets used and/or analysed during the current study are available from the corresponding author upon reasonable request.

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
