# Peer review of "Characteristics of Fitness-Related Injuries in The Netherlands: A Descriptive Epidemiological Study"

_sports, 2022, doi:10.3390/sports10120187_

Round 1
Reviewer 1 Report
First of all, I would like to congratulate the authors for the quality and clarity of the manuscript, as it describes in great detail the reality of the Netherlands in relation to injuries, which can be a point of reference for professionals in the field.
However, I have some minor comments that could enrich the manuscript.
A section on practical applications of this data could be added. How can this information be taken away and what will it be useful for professionals in the sector?
And finally, the discussion is a bit poor, there are few references to previous studies with which to discuss the results obtained. And in some cases the study is continually repeated (line 241 to 251).
Broadening the search for studies could enrich your discussion.
Author Response
Dear reviewer, thank you for your comments. we have provided a point by point response.
First of all, I would like to congratulate the authors for the quality and clarity of the manuscript, as it describes in great detail the reality of the Netherlands in relation to injuries, which can be a point of reference for professionals in the field.
Authors reply: Thank you very much for these compliments.
However, I have some minor comments that could enrich the manuscript.
A section on practical applications of this data could be added. How can this information be taken away and what will it be useful for professionals in the sector?
Authors reply: Thank you for this suggestion. We have incorporated some practical applications in the paragraph in which we address the Sequence of Prevention. See also: According to van Mechelen’s sequence of prevention [6], insight in injuries, sports activities and risk factors is necessary to ultimately develop and implement injury preventive interventions. Based on our data, which fit in the first quadrant of the sequence of prevention, some clues for prevention and practical implications can be derived. Most injuries occur during strength training and individual cardio exercises. Special attention should be given to the shoulder during almost all types of gym-based fitness activities, but at least during strength training. A practical implication might be that trainers always supervise (complex) exercises that involve the shoulder, until the trainee is capable of executing the exercise in a correct way. As more than seventy percent of the injuries (n=361, 73%) occurred during unsupervised fitness activities and at home, injury preventive interventions must be able to be carried out without supervision, and/or outside the gym, or supervision should be stimulated. Since supervision is not always possible less complex exercises should be advised to prevent the occurrence of injuries.“.
And finally, the discussion is a bit poor, there are few references to previous studies with which to discuss the results obtained. And in some cases the study is continually repeated (line 241 to 251). Broadening the search for studies could enrich your discussion.
Authors reply: During the revision process we have included other studies in the discussion section. We believe this section is less poor now.
Reviewer 2 Report
Please see the attachment.

Author Response
Dear reviewer, thank you for uoyr comments. Wwe have given a point-by-point response.
Reviewer 2
In their study titled ‘Characteristics of fitness-related injuries in the Netherlands; a descriptive epidemiological study’ the authors provide an overview of self-reported injuries sustained in gym-based exercise among people in the Netherlands. The study sample is nested in a large-scale panel regularly participating in surveys. The focus of the study is on injury type, cause and activity when injured, outcomes, and (self-reported) insights into prevention. The authors provide a solid background and justification for the study, the methods and results are well presented, and the discussion addresses the strengths and limitations.
Authors reply: Thank you very much for this positive comment.
There are a few major and minor concerns, which I will outline below.
Major concerns:
Can you try to quantify participation bias please. Only 12% of panel members who were contacted, opened the link to the survey. How did those who did differ from those who didn’t open the link (in terms of sociodemographics, and if you have this information, also in terms of exercise habits)? Survey participants sometimes ‘self-select’: they may read the study title and decide not to open the link if, for example, they never exercise.
Authors reply: We have gathered information from the market research company to discuss participation bias. We have added a section on this topic to our discussion section. See page 12, limitations section: “Finally, the online selection procedure through a respondent panel may have induced selection and participation bias. In advance, from a small sample of the panel it was known that they participated in gym-based fitness activities. The online questionnaire was sent to this small sample first. Then, a random sample, representative for the Dutch population, was contacted. At last, the sample was enriched with panel members of low or middle education level to compensate for any under-reporting of these members, The research company monitored the response from the panel members. Although the research company attempted to minimise selection bias by using a representative sample, (non-)participation bias might still occur. Non-participation bias can affect the internal validity of the results in every study. In ours, twelve percent of the contacted panel members opened the link to the screening questions. Although a click through rate of 12% was rather normal for their panel members according to the research company (addressed in personal communication with the authors), one small explanation for this percentage was that at the time of the study, the research company had one panel in which both respondents for quantitative research and for qualitative research were member. Since our study was of a quantitative nature, panel members who only want to participate in qualitative research will not have opened the link. However, this only accounts for 5-10% of the panel. Unfortunately, it was not possible to compare the non-participants with the participants in e.g. terms of exercises habits. However, the results of our study seem comparable with national results for several characteristics [4]. Hence, we believe, the effect of participation bias on our internal validity is acceptable.”.
Minor concerns:
In the title, please replace ‘;’ with ‘:’.
Authors reply: We have replaced this accordingly.
In the abstract, please include information on the study setting/sample, place and recruitment methods.
Authors reply: We have added some information on the study setting/sample, place and recruitment where possible within the boundaries of the abstract. See page 1, lines 15-18: “The survey was distributed by a market research agency. A total of 494 Dutch fitness participants aged >17 years (mean 38.9, 59% male), who sustained a fitness-related injury in the preceding 12 months were included in the study.”
In the abstract, please provide age and sex of participants.
Authors reply: We have added the mean age and proportions of males in the abstract. See also reply above.
In the abstract, the statement ‘Only 38.9 percent had returned to their normal fitness activities’ requires information on the follow-up duration.
Authors reply: We have decided to remove this sentence from the abstract. We agree with the reviewer that more information on follow-up is needed. We have discussed this statement in our discussion session; this might be to lengthy for an abstract.
Introduction line 30: please replace ‘chance’ with ‘risk’
Authors reply: We have replaced ‘chance’ with ‘risk’.
In the introduction, ‘Little is known about injuries sustained during gym-based fitness activities’ –this is not quite right; lots is known. Please be more specific about what is and what isn’t known, and which knowledge gap this study is addressing. Please also emphasis the ‘why’ – i.e. the study purpose. Which is, I think, to guide and inform prevention of gym-related injuries?
Authors reply: After reading your comment, we have removed the first sentence of this paragraph. We believe that what is known and what isn’t, is mentioned, but could however be made more clear. We have added a sentence on the ‘why’ to the paragraph as well. See page 2, paragraph above methods and materials.
In the introduction, when summarising the literature (lines 54-58), please provide context – when and where were these studies conducted? Does it matter – i.e., are the findings specific to country/setting?
Authors reply: We have added some information of the mentioned studies to the introduction section. We may conclude now that the context of the studies does not matter that much. Of course different studies and study setting will lead to some differences, but still some similarities exits. Therefore some general implications for injury prevention are addressed in the discussion section.
Methods line 71: should this be ‘Leef-stijlmonitor’?
Authors reply: The Dutch correct name is Leefstijlmonitor. We have however, now used the official English name of the survey: Health Survey/ Lifestyle monitor, as this will be more understandable for readers.
Methods line 71: I think ‘Dutch national questionnaire’ should be ‘Dutch national survey’ (or population survey).
Authors reply: Yes, you are correct, we have adjusted this accordingly.
Methods: when referring to the 55,000 panel members, can you please provide a reference with information about the representativeness of this panel?
Authors reply: In 2020, the panel we used was owned by the company Respondenten.nl. They had a website with information about their panel, but in 2021 they became part of the Norstat Group (www.norstat.nl), with panel members in 18 European countries. On the former website, amongst others, the following information was available:
- At least 55,000 panel members
- Good spread over all regions of the Netherlands
- 62% female, 38% male
- 12% low, 37% middle 51% highly educated
- 43% < 34 years, 29% 35-49 years, 28% > 50 years
Unfortunately, the information about this panel is no longer accessible. We do however, can mention the name of the research company, although we don’t know whether this is usual.
Results: please take care with use of comma vs. full-stop in numbers (e.g. 23.862 in line 106 should be 23,862).
Authors reply: Thank you for noticing this, we have checked the manuscript. We believe all numbers are correct now.
Results: how were the 23,862 out of the panel of 55,000 selected?
Authors reply: the research company have selected three groups of respondents within their panel. First adults respondents who participated in relevant sports were invited (total 3,310 invited), with a preselection on the following sports: Aerobics, Cross-fit, Dancing/ballet, Boxing, Fitness, Gymnastics, Pilates, Spinning, Yoga.
Then a group of the general Dutch population aged 18 and older was selected, with no further preselection. (total 17,741 invited). To make sure enough Dutch people who have a low of middle level education were questioned, a sample of practically educated adults were invited. (total 2,625 invited).
We have added some of this information in the results section. Furthermore, we have discussed participation bias in the discussion section.
Table 1: please replace this with a standard flow-chart.
Authors reply: We have replaced table 1 with a flow chart (figure 1).
Table 2: most journals prefer 95% confidence intervals rather than standard deviations. I would also like to see the age range.
Authors reply: We have added the age range and the minimum and maximum score. So we did for height, weight and working hours. Furthermore as we didn’t know what the journal prefers, we have contacted the editor. We now have added the SE of the mean and the 95% CI around the SE in the table.
Table 2: please provide breakdown (%) of the study sample by categories such as: (primarily) working for income; studying; unemployed; retired.
Authors reply: Unfortunately, we do not have this specific information, but we do know that 19 respondents worked less than 1 day, and 179 respondents had a fulltime job (36 hours or more per week. We have added this information to the table.
Figure 1: please provide a high-resolution figure; this is very blurry.
Authors reply: We have added a new version of this figure. We hope it is less blurry now.
Results line 182 (and onwards): please avoid the term ‘treated medically’ and replace with ‘seeking health services’ or ‘contact with health care provider’.
Authors reply: We have adjusted this in the manuscript.
Table 6 can’t be interpreted without information on the duration of follow-up: this can range (if I’ve understood correctly) from several days to 12 months. More information is provided in the paragraph below (line 188-200), which takes follow-up duration into account. I suggest replacing table 6 with the information that takes follow-up duration (in quarters) into account.
Authors reply: We have read the paragraph below table 6 and as all the information of table 6 (and more) is mentioned in this paragraph, we have removed table 6.
Discussion of recall bias (line 235-237): please provide a reference. You can also refer to ‘telescoping’.
Authors reply: We have added two references reference on this topic, and expanded the discussion section on this limitation. See page 12: “The injury should have been sustained in the preceding 12 months. Longer recall periods of e.g. 12 months, significantly underestimate the injury rate compared to shorter recall periods. Shorter recall periods (1-3 months) are recommended for injury questionnaires [22], and should be used when calculating the overall non-fatal injury incidence rate [23]. However, longer recall periods (12 months) may be safely used to obtain information on the more severe, but less frequent, injuries [23]. Furthermore, in another study on the accuracy of patient recall for self-reported doctor visits, it showed that both bias and variance were minimised for the 12‐month recall period, compared to the 2-weeks and 3-months recall period [24]. According to these authors, this result may reflect telescoping that occurs with shorter recall periods (participants pulling in important events that fall outside the period) [24]. Our 12-months recall period could have led to recall bias, as less severe injuries could be under-reported. However, as the aim of this study was to gain insight in the injury problem, and then derive first clues for prevention, we believe that our results are still useful.”.
Discussion: ‘Of the 494 fitness participants, 39 percent (n=192) had returned to their normal fitness activities at the time of this study.’ This is difficult to interpret as the follow-up duration was not uniform across the sample.
Authors reply: This is indeed true. We have added information in the results section on fully return to fitness activities related to the onset of the injury (paragraph 3.3), and discussed this in the discussion section. See also: “Of the 494 fitness participants, 39 percent (n=192) had returned to their normal fitness activities at the time of this study. More recently sustained injuries, and thus with a shorter follow-up period in this study, are probably less often fully cured than injuries sustained 10-12 months previously, and thus could impede a return to normal fitness activities. Therefore, we asked the fitness participants how long ago they sustained their injury. Analyses showed that fitness athletes who sustained an injury 10-12 months ago, did not have fully returned to sports more often than fitness athletes who sustained their injury in the preceding 0-9 months. A possible explanation might be that participants who got injured 10-12 months ago did report more severe injuries, as only 45% of them were fully recovered, compared to 55% on average.”.
Discussion lines 265 to 276: this belongs in the Results section.
Authors reply: we have removed theses lines. They were already present in the results section.
Discussion lines 292-299: excellent. Placing this in the context of exposure would be great as it
allows for calculation of rates.
Authors reply: Thank you for this compliment. In future studies hopefully we will be able to take exposure more into account.
Discussion: in the limitations section, please discuss the various biases incl. participation bias.
Authors reply: We have extended the limitation section in line with your earlier comment, and discusses selection bias and participation bias. Please see our earlier response to your comments.